# Imaging-Based Characterization of a *Slco2b1^(-/-)^* Mouse Model Using [^11^C]Erlotinib and [^99m^Tc]Mebrofenin as Probe Substrates

**DOI:** 10.3390/pharmaceutics13060918

**Published:** 2021-06-21

**Authors:** Solène Marie, Irene Hernández-Lozano, Louise Breuil, Charles Truillet, Shuiying Hu, Alex Sparreboom, Nicolas Tournier, Oliver Langer

**Affiliations:** 1Laboratoire d’Imagerie Biomédicale Multimodale, BIOMAPS, Service Hospitalier Frédéric Joliot, Université Paris-Saclay, CEA, CNRS, Inserm, 4 Place du Général Leclerc, 91401 Orsay, France; solene.marie@aphp.fr (S.M.); louise.breuil@universite-paris-saclay.fr (L.B.); charles.truillet@universite-paris-saclay.fr (C.T.); 2Département de Pharmacie Clinique, Faculté de Pharmacie, Université Paris-Saclay, 92296 Châtenay-Malabry, France; 3AP-HP, Université Paris-Saclay, Hôpital Bicêtre, Pharmacie Clinique, 94270 Le Kremlin Bicêtre, France; 4Department of Clinical Pharmacology, Medical University of Vienna, 1090 Vienna, Austria; irene.hernandezlozano@meduniwien.ac.at (I.H.-L.); oliver.langer@meduniwien.ac.at (O.L.); 5Division of Pharmaceutics and Pharmacology, College of Pharmacy, The Ohio State University, Columbus, OH 43210, USA; hu.1333@osu.edu (S.H.); sparreboom.1@osu.edu (A.S.); 6Department of Biomedical Imaging and Image-guided Therapy, Medical University of Vienna, 1090 Vienna, Austria

**Keywords:** OATP2B1, drug transporters, [^11^C]erlotinib, [^99m^Tc]mebrofenin, liver, imaging, pharmacokinetic modeling

## Abstract

Organic anion-transporting polypeptide 2B1 (OATP2B1) is co-localized with OATP1B1 and OATP1B3 in the basolateral hepatocyte membrane, where it is thought to contribute to the hepatic uptake of drugs. We characterized a novel *Slco2b1^(-/-)^* mouse model using positron emission tomography (PET) imaging with [^11^C]erlotinib (a putative OATP2B1-selective substrate) and planar scintigraphic imaging with [^99m^Tc]mebrofenin (an OATP1B1/1B3 substrate, which is not transported by OATP2B1). Dynamic 40-min scans were performed after intravenous injection of either [^11^C]erlotinib or [^99m^Tc]mebrofenin in wild-type and *Slco2b1^(-/-)^* mice. A pharmacokinetic model was used to estimate the hepatic uptake clearance (CL_1_) and the rate constants for transfer of radioactivity from the liver to the blood (*k*_2_) and excreted bile (*k*_3_). CL_1_ was significantly reduced in *Slco2b1^(-/-)^* mice for both radiotracers (*p* < 0.05), and *k*_2_ was significantly lower (*p* < 0.01) in *Slco2b1^(-/-)^* mice for [^11^C]erlotinib, but not for [^99m^Tc]mebrofenin. Our data support previous evidence that OATP transporters may contribute to the hepatic uptake of [^11^C]erlotinib. However, the decreased hepatic uptake of the OATP1B1/1B3 substrate [^99m^Tc]mebrofenin in *Slco2b1^(-/-)^* mice questions the utility of this mouse model to assess the relative contribution of OATP2B1 to the liver uptake of drugs which are substrates of multiple OATPs.

## 1. Introduction

Organic anion-transporting polypeptides (OATPs) belong to the solute carrier (SLC) superfamily of transporters and are involved in the tissue uptake of endogenous compounds and many clinically used drugs [1]. OATP2B1 (encoded by the *SLCO2B1* gene) is one important member of the OATP family and has a rather broad tissue expression profile including the small intestine, liver, kidneys, brain, heart, lungs and skeletal muscle [2,3,4,5,6]. Several drugs have been identified in vitro as substrates of OATP2B1, which is believed to be involved in the intestinal absorption of orally administered drugs. Clinically relevant OATP2B1-mediated food–drug interactions have been reported with certain fruit juices which contain OATP2B1-inhibiting components (e.g., naringin), leading to decreased systemic exposure of OATP2B1 substrate drugs (e.g., fexofenadine, aliskiren and celiprolol) [7,8,9]. OATP2B1 is co-localized with OATP1B1 (*SLCO1B1*) and OATP1B3 (*SLCO1B3*) in the basolateral/sinusoidal membrane of hepatocytes, where it is believed to mediate the hepatic uptake of drugs from the systemic circulation and thereby contribute to their hepatic clearance. However, OATP2B1 shows a largely overlapping substrate profile with OATP1B1/1B3, meaning that most drugs have a low fraction transported (f_t_) by OATP2B1 at the basolateral hepatocyte membrane (f_t_ equals the ratio of the hepatic uptake clearance of a drug by OATP2B1 to the total hepatic uptake clearance) [10]. Due to the lack of clinically validated OATP2B1-selective inhibitors, the role of OATP2B1 in the hepatic disposition of drugs has remained elusive thus far. OATP2B1 has been defined by the International Transporter Consortium as a transporter of emerging clinical importance [11]. It has been recommended that the interaction of drugs with OATP2B1 has to be evaluated when there is in vivo evidence of intestinal or hepatic transport that cannot be attributed to other well-known mechanisms [11].

A step forward in efforts to better understand the intrinsic role of OATP2B1 in drug disposition has been the recent generation of *Slco2b1^(-/-)^* mice [12,13]. A first characterization of these novel mouse models revealed significantly lower plasma concentrations of orally administered fexofenadine and fluvastatin in *Slco2b1^(-/-)^* mice as compared with wild-type mice, which supported a role of OATP2B1 in the intestinal absorption of these drugs [12,13]. In an attempt to address the role of hepatic OATP2B1, the plasma pharmacokinetics (PK) of fluvastatin were also studied after intravenous (i.v.) administration, which failed to reveal significant differences between *Slco2b1^(-/-)^* and wild-type mice [13]. Given that liver concentrations may be a better indicator than plasma concentrations of the influence of OATP2B1 on the hepatic uptake of drugs, investigators also measured fluvastatin liver concentrations in *Slco2b1^(-/-)^* and wild-type mice after oral drug administration [13]. However, neither liver concentrations nor liver-to-plasma concentration ratios differed between the two mouse strains. These measurements were only made at one single and late time point after oral drug administration (4 h) and may have therefore failed to reveal the contribution of OATP2B1 to the initial hepatic uptake of fluvastatin.

Molecular imaging methods, such as positron emission tomography (PET), single-photon emission computed tomography (SPECT) or planar scintigraphic imaging, can be used to dynamically and non-invasively measure the organ and tissue PK of radiolabeled drugs in animals and humans [14]. This approach has been termed as PK imaging and has proven suitable to assess the influence of membrane transporters on drug tissue distribution and excretion [14]. With dynamic acquisition and dedicated compartmental PK models, quantitative parameters describing drug transfer across biological membranes can be obtained, which can be related to the activity of different membrane transporters [15,16,17].

The aim of the present work was to further characterize a novel *Slco2b1^(-/-)^* mouse model [13], with an emphasis on the role of OATP2B1 in the tissue distribution of its substrates. To this end, we used PET imaging with the putative OATP2B1-selective probe substrate [^11^C]erlotinib [18,19] and planar scintigraphic imaging with the OATP1B1/1B3 probe substrate [^99m^Tc]mebrofenin [20,21,22]. A previously developed PK model was employed to assess the role of OATP2B1 in the hepatobiliary disposition of the employed radiotracers [15].

## 2. Materials and Methods

### 2.1. Radiotracers

[^11^C]erlotinib was synthetized as described previously [23]. The radiochemical purity was >98%, and the molar activity at the time of injection was 13.6 ± 7.9 GBq/µmol. Commercial kits of mebrofenin (Cholediam^®^) were a gift from Mediam (Marcq en Baroeul, France). Each kit was labeled with a sodium [^99m^Tc]pertechnetate eluate (148 MBq/mL) obtained from a sterile ^99^Mo/^99m^Tc generator (Tekcis^®^, distributed by GE Healthcare, Buc, France) followed by quality control according to the manufacturer’s recommendations.

### 2.2. Animals

Male adult wild-type (*n* = 10) and *Slco2b1^(-/-)^* (*n* = 9) mice with a C57BL/6N background were obtained via the University of California, Davis Knockout Mouse Project (KOMP) Repository (www.komp.org) as a gift from the Ohio State University (Columbus, OH, USA), where the transporter-deficient mouse model was generated as described previously [13]. At the time of the experiments, wild-type and *Slco2b1^(-/-)^* mice weighed 29.3 ± 5.0 g and 33.2 ± 2.4 g, respectively. All mice were housed in a temperature- and humidity-controlled environment with a 12-h light/dark cycle and received a standard diet and water ad libitum. All animal experiments were performed in accordance with the recommendations of the European Community (2010/63/UE) and the French National Committees (law 2013–118) for the care and use of laboratory animals. The experimental protocol was approved by a local ethics committee for animal use (CETEA) and by the French Ministry of Agriculture (APAFIS: 7466-2016110417049220 v3, accepted on 21 December 2018).

### 2.3. PET and Planar Scintigraphic Imaging

Mice were anesthetized with isoflurane (3.0% for induction, 1.5–2.5% for maintenance) in 100% oxygen, and a catheter was inserted into the caudal tail vein for i.v. radiotracer injection. PET imaging was performed using an Inveon^®^ microPET system (Siemens, Knoxville, TN, USA). Following i.v. injection of [^11^C]erlotinib (7.6 ± 1.1 MBq, corresponding to 0.36 ± 0.37 µg of unlabeled erlotinib), a dynamic PET scan was acquired over 40 min. Dynamic PET data were sorted into 20 frames with time durations of 3 × 0.5 min, 5 × 1 min, 5 × 2 min, 3 × 3 min, 3 × 4 min and 1 × 2.5 min.

[^99m^Tc]mebrofenin planar scintigraphic imaging was performed using a clinical Symbia^®^ SPECT-CT camera (Siemens) with a low-energy high-resolution (LEHR) collimator. Mice were i.v. injected with [^99m^Tc]mebrofenin (7.2 ± 0.2 MBq) followed by dynamic planar acquisitions. Dynamic images were reconstructed in 54 frames with time durations of 20 × 0.25 min, 10 × 0.5 min, 20 × 1 min and 4 × 2 min.

### 2.4. Analysis of PET Data

Volumes of interest for the left ventricle of the heart (image-derived arterial blood curve), liver, intestine (representing all the visible intestinal radioactivity), brain, myocardium, skeletal muscle (quadriceps femoris muscle), left kidney and right lung were manually outlined on the reconstructed PET images with the software AMIDE [24]. We assumed that radioactivity in the intestine predominantly represented excreted bile and that no direct secretion of [^11^C]erlotinib from blood into the intestine occurred during the short duration of the PET scan (40 min). The approach for obtaining an image-derived blood curve has been previously validated by correlating image-derived with sampled blood radioactivity concentrations [25]. Time–activity curves (TACs) were extracted for each volume of interest and were expressed in percent of injected dose per mL (%ID/mL) for all examined tissues except for the intestine, for which radioactivity was expressed as %ID (by multiplication of the image-derived radioactivity concentration with the volume of interest). For PK modeling purposes, the TACs were represented in megabecquerel per mL (MBq/mL) for the blood and liver and in MBq for the intestine.

The area under the TAC (AUC, %ID/mL x min) was calculated for each selected volume of interest using Prism Software, Version 8.4 (GraphPad, La Jolla, CA, USA). In order to assess the distribution of [^11^C]erlotinib to the brain, myocardium, skeletal muscle, liver, kidneys and lungs, the tissue-to-blood AUC ratios (AUC_tissue_/AUC_blood_) were calculated. AUC_blood_ was calculated from the image-derived blood curve from the left ventricle of the heart. In addition, the tissue uptake rate constants (*k*_uptake,tissue_, mL/min/mL tissue) of radioactivity were estimated using integration plot analysis [26] by employing the following equation:(1)$$\frac{X_{t,\mathrm{tissue}}}{C_{t,\mathrm{blood}}}=k_{\mathrm{uptake},\mathrm{tissue}}\times \frac{{\mathrm{AUC}}_{0-t,\mathrm{blood}}}{C_{t,\mathrm{blood}}}+V_{E,\mathrm{tissue}}$$
where X_t,tissue_ is the amount of radioactivity per milliliter tissue at time t, C_t,blood_ is the radioactivity concentration in the blood (image-derived blood curve) at time t, AUC_0−t,blood_ is the blood AUC from time 0 to time t and V_E,tissue_ is the y-intercept of the integration plot. *k*_uptake,tissue_ represents the slope of the early linear part of the integration plot and is estimated by performing linear regression analysis.

### 2.5. Analysis of Planar Scintigraphy Data

Images were analyzed with PMOD^®^ software (version 3.9, PMOD Technologies LLC, Zürich, Switzerland) as described in a previous study in rats [27]. Regions of interest were manually drawn over the liver, intestine (assumed to represent excreted bile) and whole heart (image-derived blood curve). Corresponding TACs were generated by plotting the mean radioactivity counts (counts per second, cps) in each region of interest normalized to the injected radioactivity amount in each animal (cps/MBq) versus time. For PK modeling purposes, the TACs were represented in cps per milliliter for the blood (by dividing the radioactivity in the whole heart region of interest by the volume of the heart adjusted for body weight obtained from the literature [28]) and in cps for the liver and the intestine.

### 2.6. Pharmacokinetic Modeling

A three-compartment model (Appendix A), modified from a previously developed model [15], was implemented to estimate the PK parameters defining the transfer of radioactivity between tissue compartments. CL_1_ (mL/min) represents the hepatic uptake clearance, and *k*_2_ (min^−1^) and *k*_3_ (min^−1^) are the rate constants describing the transfer of radioactivity from hepatocytes into blood and from hepatocytes into excreted bile, respectively. The model accounts for radiotracer delivery to the liver via both the hepatic artery and the portal vein. The radioactivity concentration in the hepatic artery was assumed to correspond to the image-derived arterial blood curve, while the concentration in the portal vein was mathematically estimated during the modeling process as previously described [15]. The final flow-weighted dual-input TAC was generated using a hepatic arterial flow fraction of 0.17 [28]. The f_t_ by OATP2B1 of [^11^C]erlotinib in the mouse liver was calculated as (CL_1,wild-type_ - CL_1, *Slco2b1*_*^(-/-)^*)/CL_1,wild-type_.

### 2.7. Statistical Analysis

Statistical analysis was performed in Graphpad Prism Software. The Shapiro–Wilk normality test was used to assess the normal distribution of the data. Differences in PK parameters between wild-type and knockout mice were assessed using the Mann–Whitney U test. The level of statistical significance was set to a *p*-value of less than 0.05. All values are given as mean ± standard deviation (SD).

## 3. Results

### 3.1. Influence of OATP2B1 on the Hepatobiliary Disposition of [^11^C]erlotinib

Wild-type and *Slco2b1^(-/-)^* mice underwent dynamic PET scans with [^11^C]erlotinib. Serial PET images of one representative wild-type mouse and one representative *Slco2b1^(-/-)^* mouse showed that, after i.v. injection of [^11^C]erlotinib, radioactivity was rapidly taken up into the liver followed by excretion into the intestine (Figure 1). Mean TACs in the blood (image-derived blood curve from the left ventricle of the heart), liver and intestine are shown in Figure 2.

Radioactivity concentrations were moderately decreased in the liver of *Slco2b1^(-/-)^* mice as compared to wild-type mice, while blood and intestinal TACs were similar in both mouse strains.

A PK model (Appendix A) was implemented to estimate the uptake clearance of radioactivity from the blood into the liver (CL_1_), as well as the rate constants for the transfer of radioactivity from the liver back to the blood (*k*_2_) and from the liver to the intestine (*k*_3_). Visually, the model provided good fits of the measured liver and intestinal TACs of [^11^C]erlotinib (Appendix A), and parameter precision (calculated as percent coefficient of variation, %CV) was acceptable (Table 1). CL_1_ and *k*_2_ were significantly decreased in *Slco2b1^(-/-)^* mice by 1.7-fold and 1.9-fold, respectively, as compared to wild-type mice, while no significant differences were observed for *k*_3_ between knockout and wild-type mice (Figure 3). The decrease in CL_1_ in *Slco2b1^(-/-)^* mice corresponds to an f_t_ by OATP2B1 of [^11^C]erlotinib in the mouse liver of 0.42.

In addition, we assessed the influence of OATP2B1 on the distribution of [^11^C]erlotinib to other organs and tissues of interest (Figure 4 and Figure 5). The liver uptake rate constant (*k*_uptake,liver_), estimated by integration plot analysis, was significantly reduced in *Slco2b1^(-/-)^* mice (Figure 4), which was in good agreement with the observed reduction in CL_1_ (Figure 3).

The myocardium and kidney uptake rate constants (*k_uptake,myocardium_* and *k_uptake,kidney_*) were significantly increased in *Slco2b1^(-/-)^* mice. The other tissue uptake rate constants (*k_uptake,brain_*, *k_uptake,muscle_* and *k_uptake,lung_*) did not differ between knockout and wild-type mice (Figure 4). Moreover, none of the tissue-to-blood AUC ratios were significantly different between *Slco2b1^(-/-)^* and wild-type mice (Figure 5).

### 3.2. Influence of OATP2B1 on the Hepatobiliary Disposition of [^99m^Tc]mebrofenin

Dynamic planar scintigraphic acquisitions were performed in wild-type and *Slco2b1^(-/-)^* mice after i.v. [^99m^Tc]mebrofenin administration. Serial CT-co-registered planar scintigraphy images of one representative wild-type mouse and one representative *Slco2b1^(-/-)^* mouse are shown in Appendix A. Mean TACs in the blood (image-derived blood curve from the whole heart), liver and intestine are shown in Figure 6. The elimination of radioactivity from the blood was delayed in *Slco2b1^(-/-)^* mice along with a moderate decrease in radioactivity in the liver and intestine.

The PK model provided good fits of the liver and intestinal TACs of [^99m^Tc]mebrofenin (Appendix A), and parameter precision was, in general, acceptable (Table 1). Model outcome parameters in wild-type and knockout mice are shown in Figure 7. CL_1_ was significantly decreased in *Slco2b1^(-/-)^* mice by 3.6-fold. There was a trend towards a decrease in *k*_2_ in *Slco2b1^(-/-)^* mice (5.5-fold decrease), but statistical significance was not reached. No significant differences were observed for *k*_3_ between knockout and wild-type mice.

## 4. Discussion

While several preclinical and clinical studies have addressed the role of OATP2B1 in mediating the intestinal absorption of drugs, the role of OATP2B1 as a hepatic uptake transporter has remained elusive thus far [10,29]. This is due to the largely overlapping substrate spectrum of OATP2B1 with the major hepatic OATP uptake transporters OATP1B1 and OATP1B3 and the lack of OATP2B1-selective inhibitors for in vivo use. A step forward in providing a better understanding of the role of OATP2B1 in drug disposition has been the development of OATP2B1-deficient mouse models (*Slco2b1^(-/-)^*) [12,13]. In the present study, we characterized the *Slco2b1^(-/-)^* mouse model initially presented by Chen et al. [13] with PET and planar scintigraphic imaging using i.v. administered radiolabeled OATP probe substrates to specifically assess the role of OATP2B1 as a hepatic uptake transporter.

The imaging data were analyzed with a dedicated liver PK model modified from a previously developed model (Appendix A) [15]. In this model, both the hepatic artery and the portal vein blood supply to the liver were considered. The flow-weighted dual blood TAC was used as an input function to the model. The radiotracer concentration in the hepatic artery was assumed to equal the radiotracer concentration derived from the left ventricle of the heart for [^11^C]erlotinib and from the whole heart region of interest for [^99m^Tc]mebrofenin (due to the limited spatial resolution of planar imaging). Since the portal vein is too small to be visualized in rodents and since the radiotracer concentration is expected to be different from that of the hepatic artery, the portal vein radiotracer concentration was mathematically estimated as previously described [30]. Although this approach has been previously validated in pigs [31], it should be noted that this validation is still needed for mice. The employed PK model assumes that no metabolism of the radiotracer occurs during the time course of the imaging scan and that the rate constants define the transfer of the unmetabolized parent radiotracer between compartments. Previous studies in mice showed that the majority (>75%) of radioactivity in plasma and the liver after i.v. injection of [^11^C]erlotinib was in the form of an unmetabolized radiotracer [25]. In addition, [^99m^Tc]mebrofenin has been shown to not undergo metabolism [32,33].

Previous in vitro data in transporter-transfected A431 cells indicated that [^11^C]erlotinib is transported by human OATP2B1, but not by human OATP1B1 and OATP1B3 [18]. However, the contribution of OATP2B1 to the uptake of [^11^C]erlotinib in OATP2B1-overexpressing A431 cells was rather small (f_t_ approximately 0.20) and only evident when low concentrations of [^11^C]erlotinib (<0.1 µmol/L) were used [18]. Studies in mice showed that the initial hepatic uptake of [^11^C]erlotinib was significantly reduced upon co-administration with a pharmacological dose of unlabeled erlotinib [34], a known OATP2B1 inhibitor [35]. In addition, the hepatic uptake rate constant of [^11^C]erlotinib was lower in mice treated with rifampicin [19], a broad-spectrum OATP inhibitor [36,37]. Similar to mice, pre-treatment with rifampicin led to a reduction in the hepatic uptake rate constant of [^11^C]erlotinib in humans, although this effect was less pronounced than in mice [19]. Altogether, the results of these previous studies suggested that OATP2B1 may partly contribute to the hepatic uptake of [^11^C]erlotinib at tracer doses. However, currently available OATP2B1 inhibitors are not specific for OATP2B1 and may interact with other transporters or drug-metabolizing enzymes [11], meaning that the specific contribution of OATP2B1 to the hepatic uptake of [^11^C]erlotinib still remains to be elucidated [19]. It is currently not known whether any other membrane transporters than OATP2B1 contribute to the liver uptake of [^11^C]erlotinib or whether it mainly occurs via passive diffusion.

The hepatic uptake clearance of [^11^C]erlotinib (CL_1_) was significantly lower in *Slco2b1^(-/-)^* mice than in wild-type mice (Figure 3), which supported the notion that murine OATP2B1 contributes to the hepatic uptake of [^11^C]erlotinib in mice. However, the absence of OATP2B1 expression did not lead to changes in the blood concentrations of [^11^C]erlotinib (Figure 2), which is consistent with [^11^C]erlotinib having only a low f_t_ by OATP2B1 (f_t_ = 0.42). Similarly, previous studies in *Slco2b1^(-/-)^* mice reported no changes in the plasma PK of the i.v. administered OATP2B1 substrate drugs fluvastatin, fexofenadine and rosuvastatin, which also suggested a low f_t_ by hepatic OATP2B1 [12,13]. This highlights the importance of measuring the dynamic liver concentrations as the lack of a hepatic uptake transporter may not cause appreciable changes in a drug’s plasma PK when f_t_ is low [10]. The reduction in CL_1_ of [^11^C]erlotinib in *Slco2b1^(-/-)^* mice (1.7-fold) was lower than previously reported in rifampicin-treated wild-type mice (2.6-fold) [19], in which a concomitant and pronounced increase in blood concentrations was observed. This indicates that apart from OATP2B1, additional rifampicin-inhibitable uptake transporters may contribute to the hepatic uptake of [^11^C]erlotinib in mice. Apart from the reduction in CL_1_, the *k*_2_ parameter, which defines the transfer of radioactivity from the liver back to the blood, was significantly decreased in *Slco2b1^(-/-)^* mice (Figure 3). Although speculative, this may point to a reduced expression of an unknown basolateral efflux transporter in *Slco2b1^(-/-)^* mice which may mediate the transfer of [^11^C]erlotinib from the liver back to the blood. The *k*_3_ parameter, which reflects biliary excretion of [^11^C]erlotinib-derived radioactivity, which has been shown to be mediated by breast cancer resistance protein (BCRP/*ABCG2*) [34,38], was unchanged in *Slco2b1^(-/-)^* mice as compared with wild-type mice (Figure 3).

Next to employing [^11^C]erlotinib as a putatively OATP2B1-selective probe substrate, we used planar scintigraphic imaging with [^99m^Tc]mebrofenin as a “negative control”, which is a probe substrate which is transported by human OATP1B1 and OATP1B3, but not by human OATP2B1 [20,21]. Biliary excretion of [^99m^Tc]mebrofenin was shown to be mediated by multidrug resistance-associated protein 2 (MRP2/*ABCC2*) and basolateral efflux from the liver into the blood by multidrug resistance-associated protein 3 (MRP3/*ABCC3*) [21]. Unexpectedly, a significant decrease in CL_1_ of [^99m^Tc]mebrofenin was observed in *Slco2b1^(-/-)^* mice (Figure 7), which was more pronounced than for [^11^C]erlotinib (Figure 3). This may either indicate that [^99m^Tc]mebrofenin is a substrate of mouse OATP2B1 (while not being transported by human OATP2B1) or that the *Slco2b1^(-/-)^* mouse model showed a reduced expression of other transporters mediating the hepatic uptake of [^99m^Tc]mebrofenin. In line with this latter assumption, Chen et al. reported decreased hepatic *Slco1b2* mRNA expression in male *Slco2b1^(-/-)^* mice, which was not observed in female animals [13]. *Slco1b2* encodes OATP1B2, which is the rodent orthologue of human OATP1B1 and 1B3. Transport of [^99m^Tc]mebrofenin by mouse OATP1B2 is supported by data in *Slco1a/1b^(-/-)^* mice, which showed a markedly decreased hepatic uptake and increased blood exposure to [^99m^Tc]mebrofenin as compared with wild-type mice [39]. Interestingly, reduced hepatic *Slco1b2* mRNA expression was not reported for the other OATP2B1-deficient mouse model developed by Medwid et al., for which the same targeting construct was used [12]. The exact reasons for the marked decrease in CL_1_ of [^99m^Tc]mebrofenin in *Slco2b1^(-/-)^* mice need further investigation but may indicate a limited relevance of the *Slco2b1^(-/-)^* mouse model to investigate the OATP2B1-mediated fraction of the liver uptake of drugs which are substrates of multiple hepatic OATPs. Interestingly, *Slco2b1^(-/-)^* mice showed a trend towards a decrease in the *k*_2_ parameter (Figure 7), which may be related to a reduced expression of hepatic MRP3 which was shown to mediate the efflux of [^99m^Tc]mebrofenin from the liver into the blood [21].

Given that OATP2B1 is, in contrast to OATP1B1 and OATP1B3, not a liver-specific OATP transporter but shows a wide tissue distribution profile, we also investigated the effect of *Slco2b1* knockout on the distribution of [^11^C]erlotinib to other tissues with known OATP2B1 expression (brain, myocardium, skeletal muscle, kidneys and lungs) [10,29]. We herein benefitted from the ability of small-animal PET to perform dynamic whole-body imaging in mice. In none of the investigated tissues, except for the liver, was a reduction in *k*_uptake,tissue_ observed in *Slco2b1^(-/-)^* mice as compared with wild-type mice (Figure 4). This suggests a negligible role of mouse OATP2B1 in mediating the uptake of [^11^C]erlotinib to other tissues than the liver. Interestingly, we observed significant increases in *k*_uptake,tissue_ of [^11^C]erlotinib in the myocardium and the kidneys, which may point to compensatory changes in other unknown uptake or efflux transporters mediating the tissue distribution of [^11^C]erlotinib in mice.

Limitations of our study include the lack of transporter protein expression data in the livers of *Slco2b1^(-/-)^* mice versus wild-type mice to support our in vivo findings as well as the limited knowledge regarding whether [^11^C]erlotinib and [^99m^Tc]mebrofenin are substrates of the rodent orthologues of human hepatic OATP transporters.

## 5. Conclusions

We performed a detailed imaging-based characterization of a novel *Slco2b1^(-/-)^* mouse model using the putative OATP2B1-selective probe substrate [^11^C]erlotinib and the OATP1B1/1B3 probe substrate [^99m^Tc]mebrofenin. Employment of a liver PK model revealed a significant decrease in the hepatic uptake clearance (CL_1_) of both radiotracers in *Slco2b1^(-/-)^* mice. This provided further support that OATP transporters may contribute to the liver uptake of [^11^C]erlotinib. However, the decreased hepatic uptake of the OATP1B1/1B3 substrate [^99m^Tc]mebrofenin in *Slco2b1^(-/-)^* mice may be related to a reduced expression of OATP1B2 (the rodent orthologue of human OATP1B1/1B3), which questions the utility of this mouse model for hepatic uptake studies with drugs which are substrates of multiple OATPs.

## Figures and Tables

**Figure 1 pharmaceutics-13-00918-f001:**
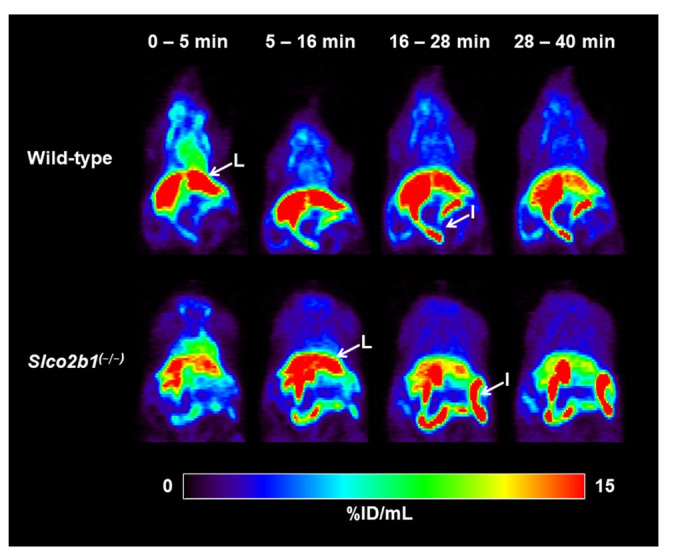
Serial PET images of one representative wild-type mouse and one representative *Slco2b1^(-/-)^* mouse after i.v. injection of [^11^C]erlotinib. Radioactivity concentration is expressed as percent of injected dose per mL (%ID/mL). Anatomical structures are labeled with white arrows (I: intestine; L: liver).

**Figure 2 pharmaceutics-13-00918-f002:**
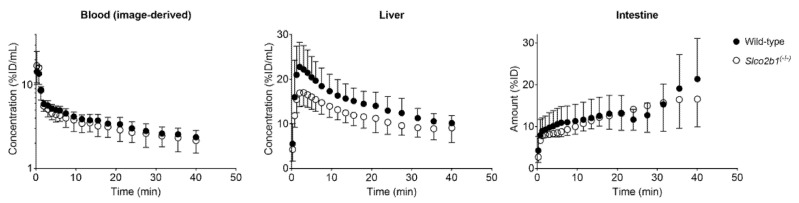
Mean time–activity curves (%ID/mL or %ID ± SD) of [^11^C]erlotinib in the blood (image-derived blood curve from the left ventricle of the heart), liver and intestine in wild-type and *Slco2b1^(-/-)^* mice.

**Figure 3 pharmaceutics-13-00918-f003:**
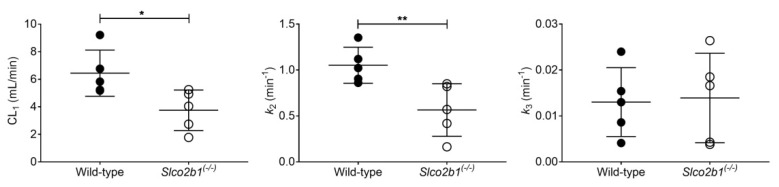
Pharmacokinetic parameters obtained with the compartmental model to describe the hepatobiliary disposition of [^11^C]erlotinib in wild-type and *Slco2b1^(-/-)^* mice. * *p* ≤ 0.05, ** *p* ≤ 0.01, Mann–Whitney U test.

**Figure 4 pharmaceutics-13-00918-f004:**
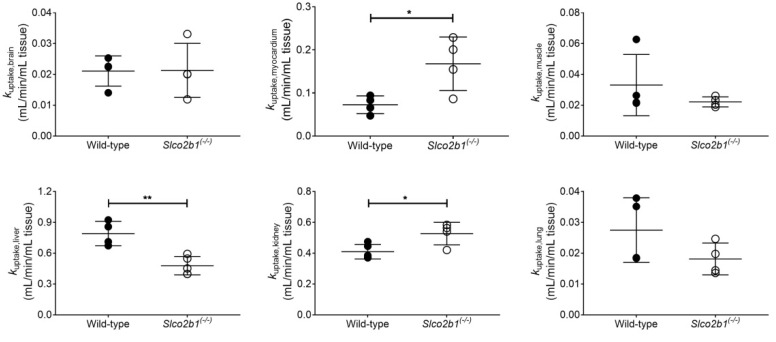
Uptake rate constants of [^11^C]erlotinib, determined with integration plot analysis, in wild-type and *Slco2b1^(-/-)^* mice for the different analyzed tissues (brain, myocardium, skeletal muscle, liver, kidney and lung). * *p* ≤ 0.05, ** *p* ≤ 0.01, Mann–Whitney U test.

**Figure 5 pharmaceutics-13-00918-f005:**
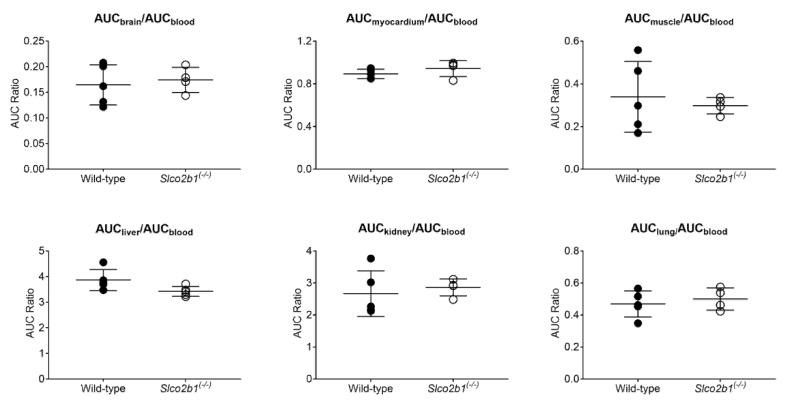
Tissue-to-blood AUC ratios of [^11^C]erlotinib in wild-type and *Slco2b1^(-/-)^* mice.

**Figure 6 pharmaceutics-13-00918-f006:**
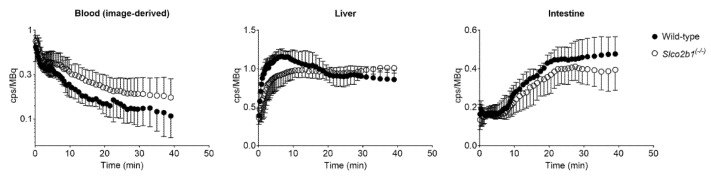
Mean time–activity curves (counts per second (cps) normalized to injected activity in MBq ± SD) of [^99m^Tc]mebrofenin in the blood (image-derived blood curve from the heart), liver and intestine in wild-type and *Slco2b1^(-/-)^* mice.

**Figure 7 pharmaceutics-13-00918-f007:**
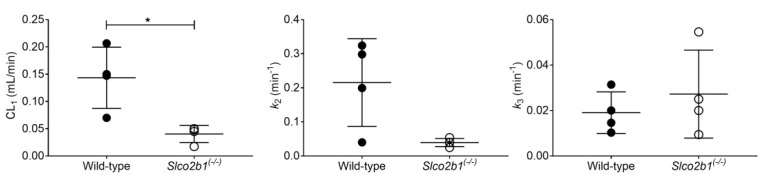
Pharmacokinetic parameters obtained with the compartmental model to describe the hepatobiliary disposition of [^99m^Tc]mebrofenin in wild-type and *Slco2b1^(-/-)^* mice. * *p* ≤ 0.05, Mann–Whitney U test.

**Table 1 pharmaceutics-13-00918-t001:** Pharmacokinetic parameters obtained with the compartmental model describing the hepatobiliary disposition of [^11^C]erlotinib and [^99m^Tc]mebrofenin in wild-type and *Slco2b1^(-/-)^* mice.

	Mice	CL_1_ (mL/min)	*k*_2_ (min^−1^)	*k*_3_ (min^−1^)
[^11^C]erlotinib	Wild-type	6.442 ± 1.675 (16.7–62.1)	1.052 ± 0.196 (15.4–54.2)	0.013 ± 0.008 (4.9–12.6)
	*Slco2b1^(-/-)^*	3.748 ± 1.470 * (5.7–57.9)	0.565 ± 0.287 ** (5.8–54.9)	0.014 ± 0.010 (3.1–10.7)
[^99m^Tc]mebrofenin	Wild-type	0.144 ± 0.056 (2.4–3.7)	0.215 ± 0.129 (2.8–6.7)	0.019 ± 0.009 (1.3–2.5)
	*Slco2b1^(-/-)^*	0.040 ± 0.016 * (4.3–43.0)	0.040 ± 0.012 (7.3–90.3)	0.027 ± 0.019 (2.3–14.4)

Data are given as mean ± SD (*n* = 4–5 per group). Values in parentheses represent the range in percent coefficient of variation (%CV). CL_1_ represents hepatic uptake clearance, and *k*_2_ and *k*_3_ are the rate constants defining the transfer of radioactivity from liver into blood and from liver to intestine, respectively. * *p* ≤ 0.05, ** *p* ≤ 0.01, Mann–Whitney U test, comparing wild-type and *Slco2b1^(-/-)^* mice.

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
