# Peer review of "Imaging-Based Characterization of a Slco2b1(-/-) Mouse Model Using [11C]Erlotinib and [99mTc]Mebrofenin as Probe Substrates"

_pharmaceutics, 2021, doi:10.3390/pharmaceutics13060918_

Round 1

Reviewer 1 Report

The authors suggested a potential utility as an OATP2B1-selective probe substrate for PET using imaging and pharmacokinetic model. Utilizing imaging in pharmacokinetic study and interpretation of its profiles may facilitate understanding of the timely dynamics for drugs and distribution into tissues. From the studies, the authors suggested that [11C]erlotinib may be used as an OATP2B1-selective probe substrate for PET. However, the results seem not enough to support the author’s claims and more concrete and detailed analysis/interpretation are necessary. Detailed comments are as follows.

  1. The authors suggested that erlotinib may be used as an OATP2B1-selective probe substrate for PET with the results showing that hepatic uptake clearance of erlotinib decreased in Slco2b1(-/-) mouse. However, CL1 for mebrofenin, which is a negative control, decreased more than erlotinib compromising the claim. Several physiological factors may be involved here, however, it is not enough to conclude for the erlotinib as a OATP2B1 probe substrate from the data presented in this study.
  2. Species differences for the transporters should be considered more extensively. It is known that OATP homolog between human and mouse is not high compared to other transporters. It is not clear that the substrates for human transporter used in this study is the substrate for murine form. The unexpected results of this study may come from the species differences of the transporters.
  3. Similar to the comment #2, expression levels of the transporters between the species should be considered as well when predicting contributions of specific transporters in total uptake of drug. If the expression levels are different between the species, the results of the mice may be not an appropriate model to predict its effect in human.
  4. It seems that the radioactivity of the drugs in intestine was assumed as a drug excreted from liver. Drugs can be distributed into the intestine even after IV infusion, more explanation is necessary for this assumption or other model should be utilized.
  5. There is no explanation for the normalization of drug amount in the blood within the tissue. It is a widely used approach to normalize blood concentration of drugs when determine tissue distribution. Even though the systemic exposure of drugs did not show statistically significant difference, we cannot completely exclude the effect of drugs contained in the blood within the tissues for determination of tissue distribution.

Author Response

Please find attached the response to Reviewer 1

Reviewer 2 Report

In this manuscript, the authors conducted PK analyses for erlotinib and mebrofenin in Slco2b1(-/-) mice using PET imaging and planar scintigraphic imaging respectively. Among OATPs, erlotinib has been reported as a selective substrate of OATP2B1, while mebrofenin is a substrate for OATP1B1/1B3. Interestingly, they found that the uptake of both drugs into the liver was reduced in the knockout mice as compared to the wild-type control. They concluded that OATP2B1 contributed to the hepatic uptake of erlotinib, while Slco2b1(-/-) mice may not be appropriate in assessing the role of OATP2B1 in the liver uptake of those compounds which are substrates of multiple OATPs. The experimental design is straightforward and the data interpretation mostly appropriate. While the authors offered potential explanation for their results in the discussion, simple experiments should have been done to explore the mechanisms. The study is rather descriptive. My specific comments are presented as the following:

  1. Erlotinib has been reported as a substrate of OATP2B1. What’s the contribution from other transporters? Their mouse results indicated that the contribution from OATP2B1 to hepatic uptake of erlotinib was rather moderate.
  2. The contribution of OATP2B1 to the uptake of erlotinib even in OATP2B1-overexpressing cells was rather small (ref 18). Why did the authors choose this compound to characterize Slco2b1(-/-) mice?
  3. It is also necessary (& important) to use the primary hepatocytes isolated from Slco2b1(-/-) mice and the wild-type control mice to validate their results.
  4. Have the authors tried with other PK models? What’s the rationale for employing the three-compartment model?
  5. Why was k2 significantly reduced in Slco2b1(-/-) mice? The authors offered a possible explanation in the discussion. But again, what are other erlotinib transporters? The authors should at least conduct an RT-PCR to examine the expression of those possible transporters.
  6. Mebrofenin was used by the authors as the “negative control” in the study. However, the results from mebrofenin were very similar to those from erlotinib. The authors discussed that it’s likely due to altered expression of other transporters for mebrofenin in Slco2b1(-/-) mice. Again, no experimental evidence was provided in this study.
  7. It is very hard to draw any solid conclusion from this study. Again, the authors should conduct further molecular characterization of transporter expression in the liver of Slco2b1(-/-) mice and perform in vitro cellular studies to support their conclusion.

Author Response

Please find attached the response to reviewer 2

Reviewer 3 Report

This is a revolutionary research apply PET and genetic mouse model to elucidate the transporter mechanisms of pharmacokinetics. The study was well organized and manuscript was well written.

Figure S3: The time-activity curves after 20 min were not fit. Is parameter estimation accurate?

I would like to see the planar scintigraphy images of [99mTc]mebrofenin. Please include representative images in supplemental files.

Author Response

Please find attached the response to reviewer 3

Round 2

Reviewer 1 Report

Accept as the revised manuscript.